# Nonuniversal impact of cholesterol on membranes mobility, curvature sensing and elasticity

Matthias Pöhl [1], Marius F. W. Trollmann [1,2] & Rainer A. Böckmann [1,2] ✉

Biological membranes, composed mainly of phospholipids and cholesterol, play a vital role as cellular barriers. They undergo localized reshaping in response to environmental cues and protein interactions, with the energetics of deformations crucial for exerting biological functions. This study investigates the non-universal role of cholesterol on the structure and elasticity of saturated and unsaturated lipid membranes. Our study uncovers a highly cooperative relationship between thermal membrane bending and local cholesterol redistribution, with cholesterol showing a strong preference for the compressed membrane leaflet. Remarkably, in unsaturated membranes, increased cholesterol mobility enhances cooperativity, resulting in membrane softening despite membrane thickening and lipid compression caused by cholesterol. These findings elucidate the intricate interplay between thermodynamic forces and local molecular interactions that govern collective properties of membranes.

Cholesterol as the most abundant lipid in plasma membranes with a total concentration of 40% in erythrocytes[1] plays a crucial role in the structure and function of cell membranes. Structurally, the steroid embeds into lipid membranes and by alignment with the phospholipids reduces the conformational freedom of their acyl chains[2]. The increased acyl chain alignment results in an increased membrane thickness and concomitantly a decreased area per phospholipid, a property called the condensing effect of cholesterol[3,4]. The different interaction of cholesterol with lipids differing in acyl chain length and saturation is one driving force in the formation of membrane phases characterized by a specific composition, packing density, and ordering[5], with cholesterol being enriched within a liquid-ordered or $L_o$ phase. The latter is frequently equated with raft domains[6]. Experimental evidence for these sterol- and sphingolipid-enriched domains is still debated, as is the sterol distribution between the leaflets of plasma membranes dramatically differing between experimental studies[7,8], with recent simulations hinting to a 28% increased cholesterol density within the extracellular leaflet[9].

With respect to function, cholesterol is involved in the formation of signaling complexes in animal cell membranes, such as for G-protein coupled receptors or immune receptors. Examples include the dimerization of chemokine receptors with cholesterol acting as a sort of molecular glue[10], or the formation of membrane domains, e.g. membrane rafts, that accompany the formation of the immunological synapse[11] and were reported to contribute to the localization of Fcγ receptors and thus immune signaling[12,13].

How does cholesterol affect the collective properties of membranes? Both in experiments[14–16] and simulations[17–19], an increase of membrane stiffness was noted upon the addition of cholesterol, in particular for bilayers containing saturated or mono-unsaturated lipids[16]. Differently, the bending elasticity for di-unsaturated 1,2-dioleoyl-*sn*-glycero-3-phosphocholine (DOPC) lipids was reported to be hardly affected or even decreased by the addition of cholesterol even up to a concentration of 50%, using electrodeformation analysis of giant vesicles[20] or X-ray scattering[16,21]. Also for sphingomyelin below the main phase transition, the bending elasticity was shown to be decreased for increased cholesterol content[20].

At variance with this nonuniversal effect of cholesterol on the bending modulus of membranes differing in acyl chain saturation[16], and combining results from neutron spin-echo experiments (NSE),

[1]Computational Biology, Department of Biology, Friedrich-Alexander-Universität Erlangen-Nürnberg, Erlangen, Germany. [2]Erlangen National High Performance Computing Center (NHR@FAU), Erlangen, Germany. ✉e-mail: rainer.boeckmann@fau.de

solid-state 2H-NMR spectroscopy (ssNMR), and molecular dynamics (MD) simulations, Chakraborty et al. reported an increase of membrane bending rigidity with increasing cholesterol concentration also for DOPC, similar to the case for saturated membranes[19]. At 50% cholesterol concentration, the bending modulus showed an even 3-fold increase compared to pure DOPC membranes.

Both ssNMR and NSE measure relaxation rates[22,23]. This temporal decay of observables was linked to time-averaged observables, here the membrane bending modulus, by making specific assumptions on membrane characteristics, including the approximation of a homogeneous membrane composition. The discrepancy to previous experiments addressing membrane bending fluctuations was explained in terms of a *hierarchical energy landscape*, suggested to result in a cholesterol-dependent increased bending rigidity only on short time and length scales[19]. I.e., at short wave length or short relaxation times, the slow lipid diffusion entails an increase in the membrane bending modulus. However, the definition of the Helfrich bending modulus does not extend to very short length scales[24]. It is also unclear how a cholesterol-induced membrane rigidification at short length scales may be reconciled with a membrane softening effect on larger length scales.

Interestingly, and in apparent contradiction to the increased stiffness of cholesterol-containing membranes or domains, bending of cholesterol-rich ordered domains ($L_o$-domain) is preferred as compared to bending of disordered, cholesterol-free domains ($L_d$-domain) with lower bending elasticity[25–27]. Cholesterol flip-flops were suggested to contribute to the deformation by relaxing tension[25,27–29]. Indeed, simulations of lipid sorting in plasma membrane tethers showed an increased concentration of cholesterol in negatively curved membrane leaflets[30] that follows its negative spontaneous curvature[9,31]. These findings rather hint towards a decreased (apparent) bending modulus at short length scales, in disagreement to the reported increase deduced from ssNMR, NSE, and MD simulations. Interestingly, a recent atomistic MD study[32] suggested a lipid-specific effect of cholesterol on bending rigidity, in agreement with electrodeformation, X-ray scattering, and tube-aspiration measurements[15].

In summary, although the influence of cholesterol on the structure of lipid membranes is well investigated and described on a molecular basis, its effect on membrane elasticity and the dependency on the degree of lipid saturation is hardly understood. Given the substantial ratio of di- and poly-unsaturated lipids and of cholesterol in plasma membranes[1], the effect of cholesterol for (local) membrane rigidity is, however, of huge biological relevance for numerous processes at the plasma membrane that necessitate membrane remodeling.

Here, we demonstrate that the impact of cholesterol on membrane properties is far from universal. Molecular dynamics simulations unveil a complex interplay between cholesterol distribution, membrane curvature, and elasticity. While cholesterol consistently induces membrane thickening and lipid condensation across various lipid membranes, it exerts contrasting effects on membrane stiffness. Specifically, fully saturated lipid membranes experience significant stiffening in the presence of cholesterol, whereas membranes composed of unsaturated lipids, such as DOPC, display only weak stiffening or even softening. This softening phenomenon can be attributed to cholesterol's negative spontaneous curvature and its increased mobility within unsaturated membranes, both laterally and transversally. These insights have profound implications for biological processes, as reduced bending elasticity facilitates localized membrane remodeling in processes like budding, endocytosis, exocytosis, and cellular signaling.

## Results
### Cholesterol enhances membrane fluctuations
The effect of cholesterol on lipid bilayers differing in headgroup chemistry and acyl chain saturation (1,2-dipalmitoyl-*sn*-glycero-3-phosphocholine, DPPC; *N*-(hexadecanoyl)-hexadecasphing-4-enine-1-

phosphocholine, DPSM; 1-palmitoyl-2-oleoyl-*sn*-glycero-3-phosphocholine, POPC; DOPC) was analyzed for lipid bilayer (short e.g. as i:DPPC) and lipid bicelle configurations (see Fig. 1a, b for DOPC bicelles, short d:DOPC) based on molecular dynamics simulations at coarse-grained (open symbols in Fig. 1) and additionally all-atom resolution (for DPPC, DOPC, and POPC, filled symbols; see Supplementary Table 1 for a list of studied systems). Similar to previous both computational and experimental studies, a condensing effect of cholesterol was observed for all studied lipid membranes, i.e. the area per lipid decreased with increasing cholesterol concentration (Fig. 1c). The increased lipid density and the packing of cholesterol molecules between the phospholipids results in an increased order of the acyl chains (exemplarily shown for all-atom DOPC model in Fig. 1b) and an increased thickness of the lipid membranes (Fig. 1d).

In addition to changes in structure, addition of cholesterol affects as well the collective properties of lipids in membranes. Here, we analyzed the effect of cholesterol on membrane thermal fluctuations in the presence and in absence of cholesterol in terms of the mean membrane curvature $\bar{H}(t)$ within a central circular disc domain (radius 3 nm, see Supplementary Methods).

Fig. 2 shows the progression and distribution of mean membrane curvatures exemplarily for symmetric monovalent fully-saturated DPPC (a,b) and doubly-unsaturated DOPC systems (c,d) in the presence (blue lines) and in the absence of cholesterol (black lines, simulation lengths 4 µs). The curvature distributions could be well fitted by Gaussian distributions centered around a vanishing total curvature, differing, however, in the distribution widths (standard deviation $\sigma$ in Fig. 2b, d). The results obtained did not differ significantly between infinite bilayers and bicelle systems (see Supplementary Figs. 4–8).

The distribution width $\sigma$ of the mean curvature values (used as a measure for the bending modulus $\kappa_b$ for bicelle systems), is slightly smaller for DOPC compared to DPPC. Addition of cholesterol led to increased fluctuations for DOPC, from 0.018 nm$^{-1}$ to 0.021 nm$^{-1}$, indicative of a softened membrane. In contrast, curvature fluctuations substantially decreased for the saturated DPPC bilayer upon the addition of cholesterol, a signature of membrane stiffening.

Thus, despite a similar ordering and condensing effect of cholesterol on all investigated types of lipid bilayers, the addition of cholesterol led to enhanced fluctuations of the mean curvature for the di-unsaturated DOPC lipid bilayer, and decreased fluctuations for fully saturated DPPC membranes.

### Cholesterol stiffening and softening of lipid membranes
To further quantify the effect of cholesterol on membrane fluctuations, we analyzed the membrane bending modulus $\kappa_b$ for all investigated systems, i.e. bicelle and infinite systems, at CG and all-atom resolution (see Supplementary Table 4). $\kappa_b$ is defined by the Helfrich Hamiltonian, which relates the membrane bending energy $E_b$ to the local principle curvatures $C_1$ and $C_2$ by an integral over the membrane surface[33,34]

$$E_b = \int dA \left\{ \frac{\kappa_b}{2} (C_1 + C_2 - C_0)^2 + \kappa_G C_1 C_2 \right\} \qquad (1)$$

The Gaussian curvature term ($\kappa_G C_1 C_2$) is disregarded in the following as it contributes a constant, the spontaneous curvature $C_0$ is zero for symmetric membranes. For infinite membrane systems, Fourier transformation of the lipid height field (FT-H) or, alternatively, of the lipid orientation field (FT-O)[35–37], and application of the equipartition theorem allows to determine bending moduli and lipid tilt moduli from fluctuation spectra[38] (see Methods Section for details) with high accuracy (Fig. 3c). For the bicelle, in turn, the local fluctuation method (LFM) in direct space was used[9]. It relates the fluctuation width $\sigma$ in mean curvature ($H = 1/2(C_1 + C_2)$) averaged over the analyzed

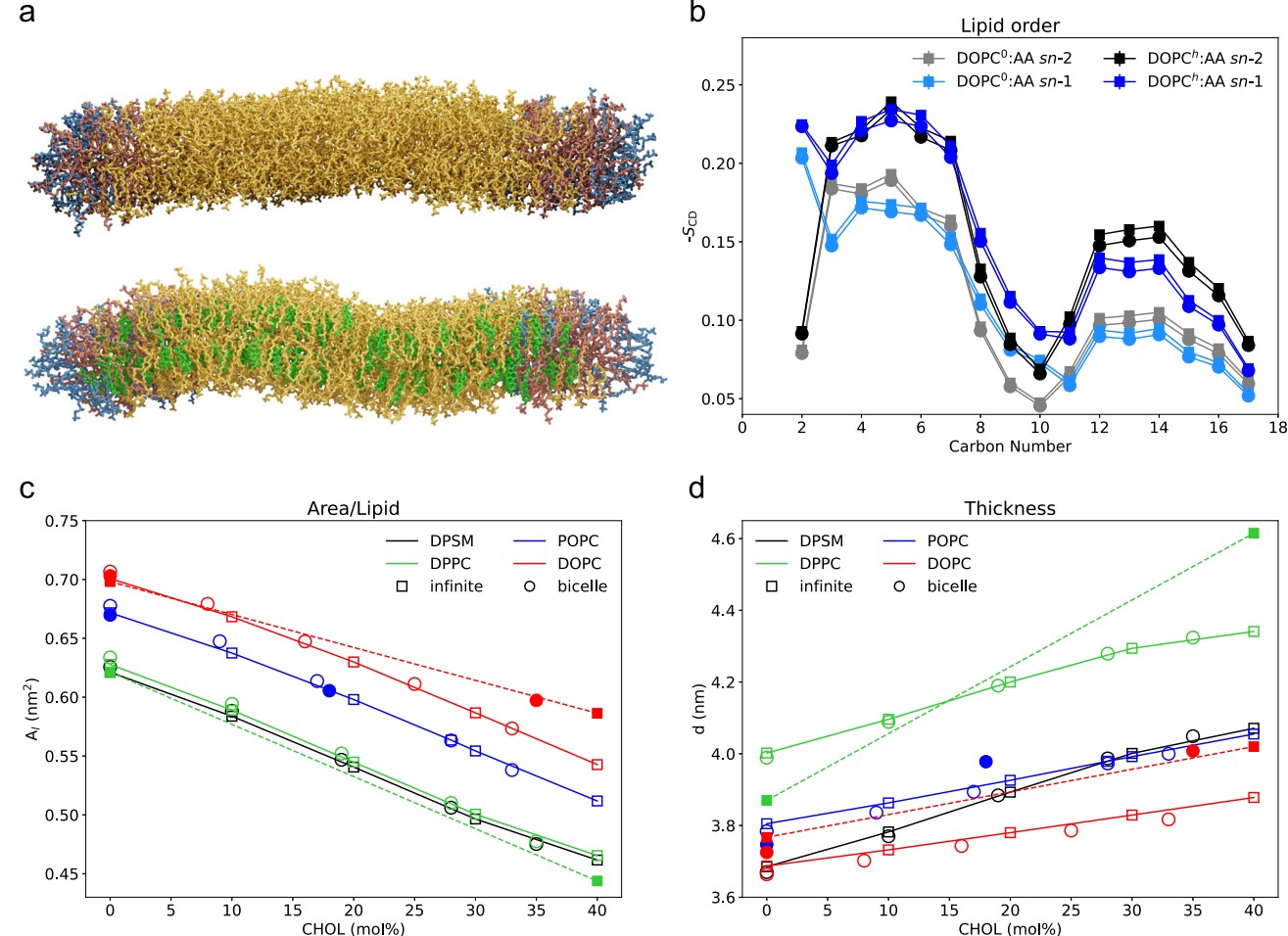

**Fig. 1 | Lipid bilayer characteristics in the presence and absence of cholesterol.** Panel (**a**) presents snapshots of all-atom (AA) DOPC bicelles, without and with cholesterol molecules. DOPC within the central bicelle domain are colored in yellow (brown in bicelle rim domain), cholesterol in green, and DTPC molecules in blue. Rendering of the images was done with Blender v3.4.1[84] and Molecular Nodes v2.4.1[85]. The study analyzes various features of lipid bilayers, including acyl chain order (**b**), area per lipid (**c**), and membrane thickness (**d**), using different setups (CG bicelles as circles and infinite, periodic bilayers as squares) and resolutions (open symbols for CG resolution and filled symbols for all-atom resolution). Error bars were smaller than the symbol size and therefore excluded.

circular disc domain to the bending modulus (see Methods Section), with $\sigma \propto \kappa^{-1/2}$.

At CG resolution (Fig. 3a), and in the absence of cholesterol, the bending modulus determined for infinite membrane systems is largest for DPPC (30.7 $k_BT$), and decreases with increasing unsaturation of the lipid acyl chains (DPSM 26.7 $k_BT$, POPC 22.1 $k_BT$, DOPC 18.8 $k_BT$; all CG simulations at 320 K). The simulation-derived values for $\kappa_b$ are in overall good agreement to experiments (see Supplementary Table 6), FT-H and FT-O methods yielded comparable results (see Supplementary Table 4). For low cholesterol concentrations (<20 mol%), the bending moduli were mildly affected and stayed approximately constant (DPPC, DPSM) or decreased by up to 5 $k_BT$ (DOPC, POPC). At increased cholesterol concentration of ≈40 mol%, the bending elasticity for DPPC strongly increased to >43 $k_BT$, i.e. the membrane substantially stiffened. In contrast, $\kappa_b$ for DOPC softened by 37% compared to the cholesterol-free case, suggesting a non-universal impact of cholesterol on membrane elasticity.

The changes in bending moduli with increasing cholesterol content overall agree between bicelles and infinite bilayer systems. However, values for $\kappa_b$ are consistently enlarged by 3–8 $k_BT$ using the LFM method as compared to values obtained from undulation and orientation spectra of infinite systems.

At all-atom resolution (Fig. 3b), the stiffening effect of cholesterol on the fully saturated DPPC membrane is substantially enhanced. At a physiological concentration of 40% cholesterol, $\kappa_b$ is almost fourfold increased compared to the cholesterol-free case (from 28.2 $k_BT$ to 106.3 $k_BT$), in good agreement with experiments for the fully saturated DMPC[14,16]. Despite a comparably high temperature of 330 K, addition of cholesterol drives the membrane into a liquid-ordered phase characterized by a significantly increased thickness (increase by 0.74 nm compared to cholesterol-free DPPC) and a very high (averaged) tail order parameter of 0.34 (Supplementary Table 3). This phase transition is insufficiently captured by the coarse-grained forcefield.

In contrast to DPPC, the bending rigidity of the di-unsaturated DOPC membrane is decreased by 31% upon the addition of 40% cholesterol at 320 K (all-atom, Fig. 3b, Supplementary Table 4), similar to the results obtained at CG resolution. However, at ambient conditions (298 K), the addition of cholesterol resulted in an increase of $\kappa_b$ from 24.1 $k_BT$ to 30.2 $k_BT$. This increase is comparable to results from a size distribution analysis of giant unilamellar vesicles (GUVs)[39].

Possible length scale-dependent effects of cholesterol-induced softening were addressed by comparing bending moduli derived from DOPC systems with lateral box lengths between 27 nm and 110 nm (Fig. 3c, CG resolution, temperature 320 K). The values of $\kappa_b$ were in excellent agreement, regardless of the size of the system. That is, cholesterol induces membrane softening in DOPC bilayers even for wavelengths beyond 0.1 μm. It is important to note that usage of standard simulation parameters for neighbourlist updates, cutoffs,

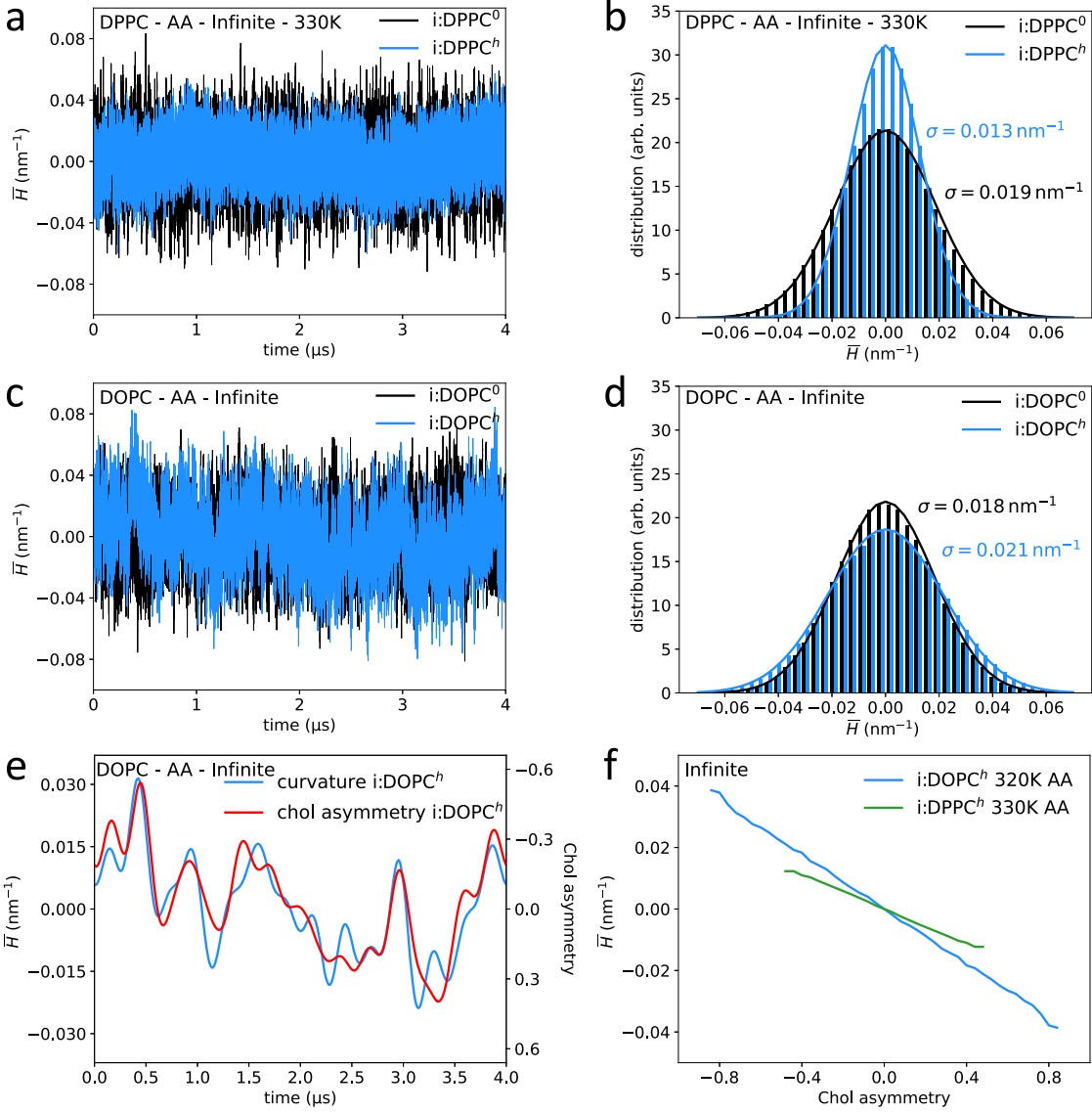

**Fig. 2 | Coupling between membrane curvature and cholesterol distribution.** Mean curvature values within circular domains of radius 3 nm were analyzed in absence (black) and presence of cholesterol (blue), separately for DPPC (at 330 K), and DOPC bilayers (320 K) at all-atom resolution. Panels **a, c** show the time development and panels **b, d** histograms for the mean curvature $\bar{H}$ with a fit assuming a Gaussian distribution. Panel **e** shows the low-pass filtered mean curvature values (blue) together with the asymmetric distribution of cholesterol between the two leaflets within the analyzed central circular domain (red; difference in the number of cholesterol molecules between the upper and lower leaflets, normalized to the average number within one leaflet). Panel (**f**) displays the time-averaged mean curvature as a function of cholesterol asymmetry for both the DPPC and DOPC bilayers.

and restraints led to imbalances in the pressure tensor[40] and artifacts in the power spectra (not shown). These issues are particularly pronounced in simulation systems exceeding >50 nm in lateral extension, resulting, for instance, in nonphysical deformations in large coarse-grained membranes[40]. In order to prevent potential biases in our spectra for both the large DOPC and DOPC/cholesterol membranes, we opted for more conservative simulation parameters, which were carefully selected and validated (Fig. 3c, Supplementary Fig. 10).

In contrast to the non-universal effect of cholesterol on membrane rigidity, an increase of the lipid tilt modulus ($\kappa_\theta$) was observed for all studied lipid bilayer systems upon cholesterol addition (Supplementary Table 4), in line with the lipid condensing effect of cholesterol and the increased acyl chain ordering (Fig. 1).

## Cholesterol and curvature are mutually dependent

In all studied systems, the cholesterol distribution to the membrane leaflets strongly correlated with the spontaneous curvature as

exemplarily shown for the all-atom DOPC bilayer system (4 μs length) in Fig. 2e employing low-pass filtered trajectories (see Supplementary Figs 7, 8 for results of DOPC simulations at CG resolution and for bicelle systems). The appearance of temporary membrane curvature within the analyzed circular domain (radius 3 nm) implies a partial redistribution of cholesterol to the respective compressed, negatively curved leaflet. Thus, cholesterol molecules preferentially partition to regions of negative curvature. The dependency between the mean curvature and the cholesterol asymmetry is approximately linear for both all-atom and coarse-grained systems (see Fig. 2f and Supplementary Fig. 9). Between the different lipid types, it is significantly enhanced for lipids with unsaturated acyl chains (Fig. 2f). Locally, the cholesterol asymmetry may even exceed 80% for high (local) curvature (shown for i:DOPC$^h$:AA at 320 K, Fig. 2f and Supplementary Fig. 11). The highly dynamic coupling between cholesterol asymmetry and the local mean membrane curvature is schematically displayed in Fig. 4 for two snapshots of a DOPC all-atom bicelle: The number of cholesterol

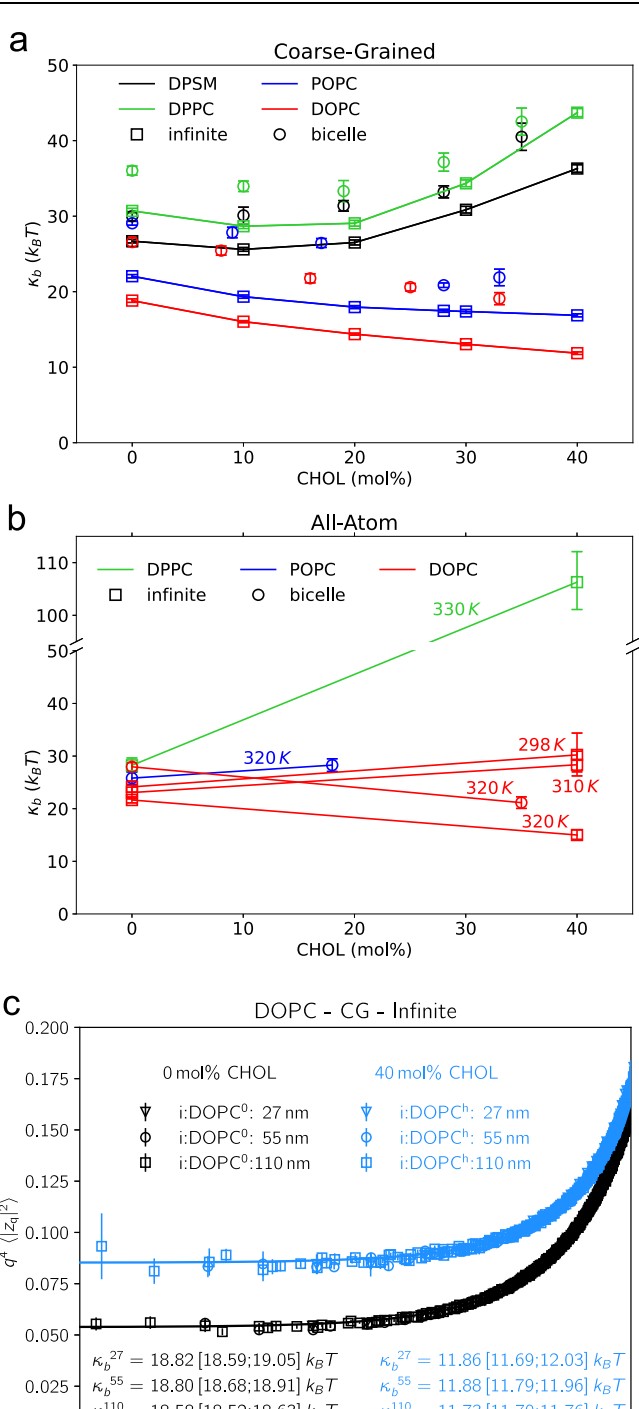

**Fig. 3 | Effect of cholesterol on membrane bending modulus.** Membrane bending moduli were determined from CG MD simulations (**a**) and from all-atom MD simulations (**b**) for different system setups (bicelles as circles and infinite, periodic bilayers as squares). The bending moduli were calculated using undulation analysis (for infinite systems) and the local fluctuation method (for bicelles). Panel **c** shows exemplarily the spectrum for DOPC bilayer systems with 0% and 40% cholesterol for different lateral box sizes (27 nm, 55 nm, and 110 nm; $\langle |z_{\mathbf{q}}|^2 \rangle$ are the amplitudes of the Fourier-transformed membrane surface). Displayed are mean values and errors as 95% confidence intervals employing parametric bootstrapping ($N = 50,000$ statistically independent samples) assuming Gaussian distributions of the mode-dependent amplitudes[78] (infinite bilayers) or standard errors of the mean employing block averaging (at least $N = 12$ independent blocks; bicelles).

molecules between upper and lower membrane leaflet within a circular domain of 7 nm radius changed by up to 50% between the snapshots separated by 1.6 μs. This change was connected to a change in mean curvature by 0.034 nm$^{-1}$ between the snapshots.

But how does the presence of cholesterol differentially affect membrane elasticity depending on lipid chain saturation? Cholesterol molecules are free to flip between the lipid leaflets of a bicelle and of periodic infinite membranes. The observed cholesterol flipping rates in the coarse-grained simulations were comparable for bicelles and periodic bilayers but differed between the investigated lipid types: Larger cholesterol flipping rates were observed for lipid membranes with a higher degree of acyl chain unsaturation, i.e. the cholesterol flipping rate increases in the order DPPC < POPC < DOPC, ranging between 0.4 μs$^{-1}$ for the fully saturated DPPC and 5.0 μs$^{-1}$ for the di-unsaturated DOPC (bicelle at high cholesterol concentration, see Supplementary Table 5). In addition to the increased rate of cholesterol flip-flops, the higher degree of acyl chain unsaturation as well results in increased lateral diffusion coefficients for cholesterol (see Supplementary Tables 2 and 3). The lateral cholesterol diffusion coefficient increases from $(0.22 \pm 0.01) \times 10^{-6}$ cm$^2$s$^{-1}$ for DPPC to $(0.48 \pm 0.01) \times 10^{-6}$ cm$^2$s$^{-1}$ for DOPC (CG resolution, at high cholesterol concentration), and, at all-atom resolution, from $(0.09 \pm 0.01) \times 10^{-6}$ cm$^2$s$^{-1}$ for DPPC (330 K) to $(0.15 \pm 0.01) \times 10^{-6}$ cm$^2$s$^{-1}$ for DOPC (320 K).

Our results suggest that the subtle alterations in membrane elasticity observed in certain experiments with di-unsaturated DOPC following cholesterol addition[16,20,41–43], in contrast to experiments and simulations on mono-unsaturated or fully saturated membranes (see Supplementary Table 6), can be attributed to an accelerated flipping of cholesterol molecules between leaflets and an increased cholesterol lateral diffusion coefficient within the leaflets. These dynamics are oriented towards elevating cholesterol concentration within negatively curved domains of lipid membranes. This dynamic redistribution of cholesterol, favoring locally enhanced curvatures, may potentially outweigh the anticipated stiffening effects resulting from cholesterol-induced lipid condensing and membrane thickening, as evidenced in the case of DOPC at an elevated temperature of 320 K.

This finding is additionally supported by results from MD simulations with artificially restricted mobility of cholesterol molecules: Blocked cholesterol flip-flops and restricted lateral diffusion of cholesterol led to a substantial increase of the bending moduli for both DPPC and DOPC lipid membranes (see Fig. 5 and Methods Section for methodological details). When the dynamics of cholesterol was restricted, the DOPC bending modulus increased from $\kappa_b = 12.0$ $k_BT$ (free cholesterol dynamics) to $\kappa_b^{noflip} = 15.9$ $k_BT$ (inhibited flipping of cholesterol) and $\kappa_b^{restr.} = 26.4$ $k_BT$ (no flipping & restrained lateral diffusion). Similarly, the membrane stiffness of cholesterol-containing DPPC membranes was substantially increased for restrained cholesterol dynamics (from 43.0 $k_BT$ to 71.6 $k_BT$). These results unambiguously show that both membranes, saturated and un-saturated, are stiffened by the structural integration of cholesterol and substantially softened by cholesterol dynamics.

## Discussion

Here, we addressed the intricate interplay of cholesterol asymmetry and membrane curvature, elucidating the impact of dynamic cholesterol redistribution on the membrane bending modulus $\kappa_b$. Our simulations uncovered apparent contrasting effect of cholesterol on membrane properties. Although cholesterol consistently induces membrane thickening and lipid condensing irrespective of lipid acyl chain saturation, cholesterol exerts both a membrane stiffening and softening: A substantial stiffening was observed for fully saturated DPPC and DPSM membranes. In contrast, the stiffening was more moderate in mono-unsaturated POPC and di-unsaturated DOPC

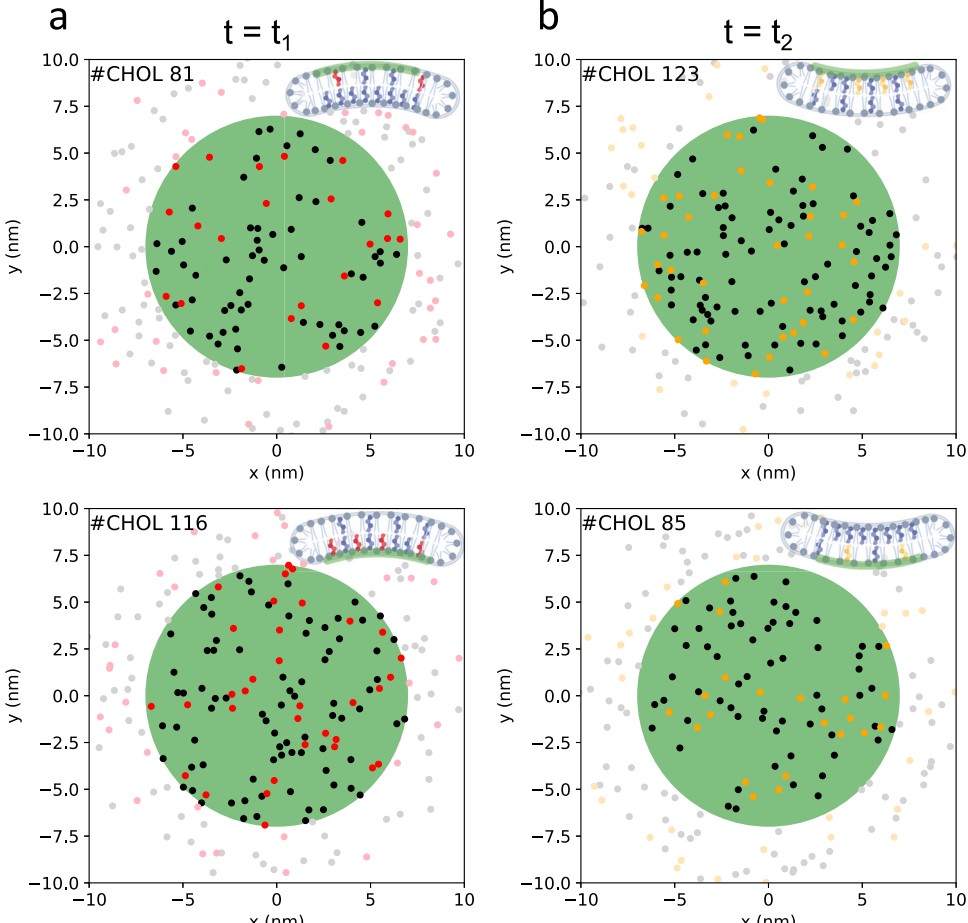

**Fig. 4 | Curvature-dependent cholesterol dynamics and distribution.** The sketch depicts the lateral positions of cholesterol molecules within the upper and lower leaflets (upper and lower panels, respectively) of a DOPC bicelle at atomistic resolution. Snapshots at distinct times, denoted as positive curvature (**a**, at time $t_1$) and negative curvature (**b**, at time $t_2 > t_1$), showcase the dynamic redistribution of cholesterol. The central domain, highlighted in green with a radius of 7 nm, serves as a reference. Cholesterol molecules departing from the central domain of the respective leaflet between these two snapshots are depicted in red, while molecules moving into the central domain of the upper/lower leaflet are represented in orange. The inset figures were created with BioRender.com.

membranes at ambient temperature (all-atom resolution). Intriguingly, this effect reversed at an elevated temperature of 320 K, at both coarse-grained and all-atom resolution, revealing a significant cholesterol-induced softening of DOPC membranes.

The (local) cholesterol distribution between the membrane leaflets was observed to be strongly coupled to fluctuations of the local membrane curvature, closely mirroring the local curvature on a sub-microsecond to microsecond timescale (Fig. 2e, f, Supplementary Fig. 11). The preference of cholesterol for membrane regions with negative curvature is in line with previous observations in experiments[44] and simulations[9,30,31]. Notably, the coupling strength between local membrane curvature and cholesterol redistribution was markedly heightened in di-unsaturated DOPC membranes compared to fully saturated DPPC membranes (Fig. 2f). This heightened coupling correlates with cholesterol mobility, encompassing both lateral and transversal movements through flip-flops, the latter revealing a nearly tenfold increase in flipping in DOPC compared to DPPC membranes. Interestingly, recent research on giant unilamellar vesicles suggested that differences in lipid spontaneous curvature connected to lateral lipid diffusion could lead to membrane softening[45]. Our results show that a similar diffusional softening[45], here by cholesterol, may even outweigh the effects of cholesterol-induced condensation and membrane thickening on membrane elasticity.

Our findings on the effect of cholesterol on the membrane bending modulus align well with earlier experimental studies directly assessing membrane elasticity. These studies, including lipids sorting to nanotubes[41], fluctuation analysis, vesicle electrodeformation[20], and X-ray scattering on bilayer stacks[21], consistently reported either unchanged or slightly decreased or increased bending elasticity of DOPC upon cholesterol addition (see Supplementary Table 6 for a comparative overview). However, recent ssNMR and NSE experiments, along with MD simulations, have introduced a contrasting and controversially discussed viewpoint by suggesting a notable up to threefold increase in the DOPC bending rigidity following cholesterol addition[19]. Here, we demonstrate that increased lipid order parameters do not necessarily indicate an increase in membrane stiffness, as assumed in ssNMR analysis based on the assumption of a homogeneous membrane model. Our simulations show an increase in lipid order parameters but a simultaneous only weak increase (at 298 K) or even decrease (at 320 K) in the bending modulus when cholesterol is added to DOPC.

Noteworthy, similar to ssNMR and NSE, a 2–3 fold increase in DOPC bending moduli upon the addition of cholesterol was as well derived from all-atom MD simulations[19]. However, the real space fluctuation (RSF) analysis heavily overestimates the bending moduli for cholesterol-rich membranes (see Supplementary Table 4). RSF deduces the bending modulus from the local splay of lipid pairs separated by no more than 1 nm. The method was shown to produce results that were in reasonable agreement with Fourier-based methods and experiments for single-component membranes, with the largest deviation observed for the doubly-unsaturated DOPC (35%)[17]. The increase in deviation from

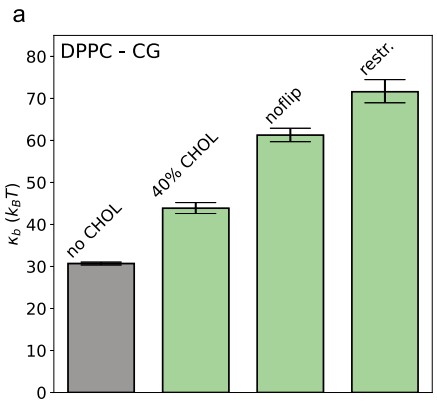

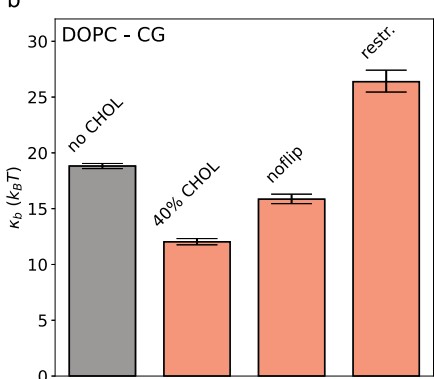

**Fig. 5 | Effect of cholesterol dynamics on membrane bending modulus.** The bending modulus was analyzed for DPPC (**a**) and DOPC bilayers (**b**) based on unperturbed coarse-grained MD simulations in the absence of cholesterol (gray bars), in presence of 40% cholesterol, with blocked cholesterol flipping between the layers (noflip), and for simulations with blocked cholesterol flipping and restrained lateral cholesterol diffusion (restr.). Displayed are mean values and errors as 95% confidence intervals employing parametric bootstrapping ($N = 50,000$ statistically independent samples) assuming Gaussian distributions of the mode-dependent amplitudes[78].

Fourier-based approaches with increasing cholesterol concentration suggests that analysis of local structure alone may not grasp the effect of diffusional softening. Differently, and consistent with our results, a recent all-atom MD study[32] showed a moderate increase of bending moduli for POPC and DOPC membranes at 300 K upon cholesterol addition (30%), by application of external forces to enforce a sinusoidal shape across periodic membrane systems combined with an umbrella sampling protocol[46]. However, the study concludes that (i) membrane stiffening is not affected by cholesterol on length scales below 10 nm and (ii) did not observe and, therefore, excludes local cholesterol enrichment as a cause of membrane stiffening. Our analysis of local mean curvature distributions for circular regions (radius 3 nm) unambiguously reveals that cholesterol addition does have an effect on curvature generation and membrane rigidity also on small length scales (Fig. 2a-d), and that curvature generation and leaflet cholesterol asymmetry are tightly interconnected (up to 80% cholesterol asymmetry in circular areas of radius 3 nm, Fig. 2f; Supplementary Fig. 11).

The simulation data implies that cholesterol adopts multiple roles in the mechanics of cellular membranes: Its lipid condensing effect stabilizes plasma membranes with respect to permeability[47] or membrane rupture[48]. This effect is pronounced when cholesterol induces a phase transition toward the liquid-ordered phase. In turn, the mobility of cholesterol in unsaturated membranes together with its spontaneous negative curvature counteracts the stiffening induced by lipid condensation and maintains a low or even reduced bending rigidity compared to that of cholesterol-free membranes. Yet, the effect of cholesterol on the mechanical properties of plasma membranes remains enigmatic, given its asymmetric lipid distribution with twofold more unsaturated and more mobile lipids in the cytoplasmic leaflet, as reported for red blood cell (RBC) membranes[1,49], and likely heightened cholesterol density in the exoplasmic leaflet[9]. Intriguingly, recent studies indicate a remarkably low bending modulus for cholesterol-rich RBC membranes of only 4–6 $k_BT$[50], contrasting with an anomalous stiffening reported for cholesterol-free model membranes with asymmetric lipid compositions[51,52]. Taken together, cholesterol likely contributes to reconciling the seemingly conflicting roles of plasma membranes, i.e., creating an effective barrier against small molecules and ions while concurrently ensuring the elasticity and plasticity essential for processes requiring dynamic membrane remodeling.

Our findings propose that the induction of local curvature by transmembrane proteins or membrane-associated proteins is facilitated by the local redistribution of cholesterol, effectively alleviating membrane stress. Notably, we observed an asymmetric distribution of cholesterol between membrane leaflets in the proximity of a curvature-inducing voltage-sensing potassium channel[9]. This effect is possibly amplified by the reported enrichment of polyunsaturated lipids near membrane proteins[53,54] that allow enhanced cholesterol diffusion and flip-flop rates.

Intriguingly, cholesterol depletion was found to shift the activation pressure of the curvature-sensing PIEZO1 channel to higher values[55]. Given that PIEZO1 activation involves substantial local membrane deformation, transitioning from a highly curved conformation to a flattened structure[56], cholesterol depletion and the resultant possibly increased membrane rigidity may heighten the energetic cost of such transitions. Furthermore, the negative spontaneous curvature of cholesterol, along with its impact on membrane rigidity, is anticipated to play a pivotal role in membrane fusion and fission events, i.e. during endo- and exocytosis. For instance, cholesterol has been identified as a pathway switch for influenza virus infection of cells by lowering the energy required for forming a hemifusion stalk[48]. Similarly, cholesterol depletion has been reported to impede curvature generation in clathrin-mediated endocytosis[57]. These insights underscore the intricate interplay between cholesterol, local membrane curvature and membrane rigidity, and membrane remodeling processes, shedding light on the impact of local membrane heterogeneity related to cholesterol mobility and its negative spontaneous curvature on collective membrane properties in cellular events.

In summary, our simulation study reflects similar to earlier experiments and simulations a condensing effect of cholesterol on membranes that is independent of the particular lipid composition. Differently, cholesterol has a non-universal effect on the stiffness of membranes: It leads to a substantial increase of bending rigidity for saturated lipid membranes, but only weak stiffening or even softening for membranes composed of unsaturated lipids such as DOPC. The observed softening could be traced back to the increased cholesterol mobility in unsaturated membranes, both lateral and transversal. The cholesterol-induced changed membrane mechanical properties have far-reaching implications for biological processes, the reduced bending elasticity will locally ease membrane remodeling processes involved e.g. in budding, endo- or exocytosis, and signaling.

## Methods
### Molecular dynamics simulations
Several different lipid types were studied at different cholesterol concentrations in a lipid bicelle setup (denoted as e.g. *d:DPPC*) and in infinite membrane systems (denoted as e.g. *i:DPPC*). The chosen bicelle setup makes use of short-chained 1,2-ditridecanoyl-*sn*-glycero-3-phospholine (DTPC) lipids and flat-bottomed potentials to reduce the

line tension, vesiculation and free diffusion across the bicelle rim, issues usually associated with bicelle systems[9]. Importantly, the bicelle model is characterized by largely unbiased thermal fluctuations as opposed to standard infinite bilayer systems[9] (see Supplementary Figs 2–8 for comparisons of mean curvatures and cholesterol asymmetries for bicelles and infinite bilayers). All systems were simulated using the GROMACS 2020(2021, 2023) simulation package[58] at coarse-grained (CG) resolution employing the MARTINI force field (version 2.2)[59–61] (see Supplementary Fig. 1 for studied lipids in MARTINI representation), and, for selected systems, additionally at all-atom (AA) resolution using the CHARMM36 force field[62]. Details of the system setup are provided below and in Pöhnl et al.[9]. Details on the different lipid compositions and simulation lengths for bicelle systems and infinite lipid bilayer systems are provided in Supplementary Table 1.

### Setup of simulation system and simulation parameters

**Bicelle systems.** The bicelle systems consist of a rim region and a central region with defined lipid composition to investigate membrane characteristics. First, a circular membrane patch of ≈9.2 nm was setup based on an initial hexagonal lipid bilayer constructed with *insane*[63], subsequent solvation in a larger rectanglular box and simulation for 20 ns with position restraints (force constant of 1,000 kJ/mol/nm²) in z-direction on the phosphate beads. Next, two rings of rim lipids (≈380 DOPC; ≈370 DTPC) were placed around the circular membrane patch, resolvated (≈120,000 CG water molecules), and equilibrated for 20 ns with position restraints on lipids of the circular membrane patch. For production simulations, cylindrical flat-bottomed potentials were applied on the phosphate beads to restrict the motion of lipids within the central region to a circular domain of radius 10.5 nm, and to restrain rim lipids to the region beyond a radius of 8.1 nm (force constant 500 kJ/mol/nm²). All systems were studied at CG resolution using the MARTINI force field[59–61] with in total ≈135,000 CG beads and a lateral box size of ≈ 32 nm. The temperature was kept at 320 K via the v-rescale algorithm[64], and the Parrinello-Rahman algorithm[65,66] with a compressibility of $3\cdot10^{-4}$ bar$^{-1}$ was used for semi-isotropic pressure coupling (isotropic for bicelle systems). A reaction field algorithm with a cut-off of 1.1 nm and a relative dielectric constant of 15 was used for electrostatic interactions. Lennard-Jones interactions were treated with a single cut-off of 1.1 nm and the time step was 20 fs.

For selected systems, all-atom simulations were performed using the CHARMM36[62] force field. Initial structures were obtained by transferring equilibrated and approximately flat bicelle structures to atomistic resolution using the *initram* protocol[67]. Resulting systems had ≈1 Mio atoms. For bicelle systems, flat-bottomed potentials were applied on the phosphate atoms of the obtained stuctures using the same radii and force constants as in the coarse-grained systems. The temperature was kept at 320 K via the v-rescale algorithm[64]. Isotropic pressure coupling to 1 bar was applied using the Parrinello-Rahman algorithm[65,66] with a compressibility of $4.5 \cdot 10^{-5}$ bar$^{-1}$. Van der Waals forces were smoothly switched to zero (between 1.0 nm and 1.2 nm) and the electrostatic interactions were treated with the PME method[68]. The time step was 2 fs. The all-atom systems were studied at a salt concentration of 0.15 M NaCl.

**Infinite membrane systems.** Infinite lipid bilayer systems at CG resolution were created with *insane*[63], further detailed in[9]. The typical lateral box size was ≈27 nm, the membranes were solvated with ≈75,000 CG water beads, resulting in systems with ≈100,000 CG beads. DOPC membranes at 0% and 40% cholesterol were additionally studied for four-fold and 16-fold increased lateral sizes, the box sizes were ≈55 nm and ≈110 nm, respectively. In order to avoid possible systematic imbalances in the pressure tensor due to missed long-range attractive van der Waals interactions as reported recently [40], we chose different settings for neighbour list updates (update every 10 steps instead of 20), an increased outer cutoff radius of $r_l = 1.5$ nm, as well as modified settings for the LINCS algorithm to solve bonded

constraints (lincs_order = 12, lincs_iter = 2). These constraint settings were shown to avoid temperature gradients for cholesterol-containing membranes[69]. We observed that both the standard neighbour list settings as well as the default settings for LINCS (lincs_order = 4, lincs_iter = 1) result in artificial membrane deformations for very large systems (>50 nm) that are reflected in artificially enhanced undulations for low $q$ values / large wavelengths (not shown).

Infinite all-atom DOPC (DPPC) systems at 0% and 40% cholesterol at a temperature of 320 K (330 K) were setup using either back-mapping or the CHARMM-GUI[70,71] (lateral box sizes between 25 nm and 28 nm, ≈900,000 atoms, ≈170,000 water molecules). DOPC membranes were additionally studied at temperatures of 310 K and 298 K (0% and 40% cholesterol, lateral box size ≈17 nm, ≈300,000 atoms, ≈60,000 water molecules).

**Restrained cholesterol dynamics.** The role of cholesterol mobility and distribution for membrane elasticity was additionally addressed for infinite lipid bilayers (lateral box size ≈18 nm, ≈33,000 CG beads, ≈20,000 water beads) at CG resolution with blocked cholesterol flipping and restricted lateral cholesterol diffusion. Starting from a homogeneous cholesterol distribution of 40 mol% in either DOPC or DPPC lipid bilayers, the cholesterol motion was restricted in lateral direction around the initial position by cylindrical flat-bottomed potentials with a radius of 0.5 nm (force constant of 5,000 kJ/mol/nm²). Cholesterol flips were prevented by adding a short-ranged repulsive interaction between the ROH bead in cholesterol and the last beads of the lipid fatty acyl chains (Lennard-Jones parameters $c_6 = 0$ kJ mol$^{-1}$nm$^6$; $c_{12} = 1$ kJ mol$^{-1}$nm$^6$).

### Analysis

Custom analysis code was written using the MDAnalysis library[72,73]. Membrane properties of lipid-bicelle systems were analyzed on the central bicelle domain (radius 7 nm). For infinite systems, the whole system was considered. Unless otherwise specified, errors were estimated using block averaging on the equilibrated part of the trajectory. The number of independent blocks was determined by monitoring the convergence of the standard error. Both for coarse-grained and all-atom simulations at least several microseconds were used for the analysis of all properties. Equilibration times were chosen such that any drift in the area per lipid, thickness, and order parameters vanished. The membrane properties are summarized in Fig. 1, and in Supplementary Tables 2 and 3.

### Membrane surface

The PO4 beads (P atoms for atomistic systems) were used to calculate the surfaces of the membrane leaflets. Surfaces of the upper $z^u(x, y)$ and lower $z^l(x, y)$ leaflets were defined as height values on a 2D lateral grid (see also[74,75]). Grid cells $(i, j)$ get assigned the weighted sum of the z-positions of surrounding PO4 beads (see Supplementary Fig. 1) as $z_{ij}$-value, using Gaussian weights $w = \exp\left(-d_{ij}^2/2\sigma^2\right)$ with $d_{ij}$ as the lateral distance between the grid point $(i, j)$ and the PO4 bead. The standard deviation was chosen to $\sigma = 0.8$ nm and the grid spacing to $\Delta x = \Delta y = 0.5$ nm. The bilayer surface was defined as the mean of the monolayer surfaces $z_{ij} = (z^u + z^l)/2$ and the local membrane normals by $N_{ij}^{u/l} = \pm\left(-\partial_x z_{ij}^{u/l}, -\partial_y z_{ij}^{u/l}, 1\right)$.

### Local membrane curvature

The bilayer surface $z_{ij}$ was used to calculate the local membrane curvature:

$$H_{ij} = \frac{\left(1 + \left(\partial_x z_{ij}\right)^2\right)\partial_y^2 z_{ij} - 2\partial_x z_{ij}\partial_y z_{ij}\partial_x\partial_y z_{ij} + \left(1 + \left(\partial_y z_{ij}\right)^2\right)\partial_x^2 z_{ij}}{2\sqrt{1 + \left(\partial_x z_{ij}\right)^2 + \left(\partial_y z_{ij}\right)^2}^3}$$

(2)

Local curvature values $H_{ij}(t)$ were averaged ($\bar{H}(t)$) over all points $i, j$ within a circular region around the bicelle center with radius $R$. For infinite systems, a similar circular region was used to calculate $\bar{H}(t)$.

## Order parameter

For coarse-grained systems the order parameter

$$P_2 = \frac{1}{2}\left\langle 3\cos^2(\theta) - 1 \right\rangle \tag{3}$$

was calculated for bond vectors in the lipid tail. Here $\theta$ is the angle between the bond vector and the local membrane normal in the upper, respectively lower, leaflet $N_{ij}^{u/l}$. The local normals were assigned from the laterally closest grid cell to the center of mass of the C1A and C1B beads (MARTINI nomenclature, see Supplementary Fig. 1). The expectation value was taken over all lipids and all tail bonds.

For all-atom systems, the order parameter $-S_{CD} = \frac{1}{2}\left\langle 3\cos^2(\theta) - 1 \right\rangle$ was calculated for all carbon atoms of each chain. Here $\theta$ is the angle between the local membrane normal and the C-H bond vectors. The local normal was calculated based on all lipids within a distance of 1.8 nm from the respective lipid. Order parameters for all investigated systems are provided in Supplementary Tables 2 and 3.

## Membrane thickness

The thickness was calculated for each lipid position within the central analysis domain and separately for both membrane leaflets. PO4 MARTINI beads (P atoms for AA simulations) were taken as lipids positions. First, the local normal was determined based on lipids within 1.8 nm distance of the analysis lipid. The (local) thickness was calculated as the distance along the local membrane normal between the analysis lipid and the averaged (projected) positions of lipids of the opposing leaflet within a distance of 5.5 nm of the analysis lipid. Values provided in Supplementary Tables 2 and 3 are averages over all analyzed frames and all lipids.

## Area per lipid

Local Voronoi tesselation was performed for each lipid (see also[76]). The lipid positions were defined as the centers of mass of the GL1 and GL2 beads (AM1 + AM2 for DPSM; ROH for CHOL; glycerol oxygen atoms for lipids and hydroxyl oxygen atom for cholesterol in AA systems). For each lipid, lipid positions closer than 4 nm were considered. The area per lipid was calculated as the area of the central Voronoi cell perpendicular to the local membrane normal and averaged over all analyzed frames and lipids (see Supplementary Tables 2 and 3).

## Lipid diffusion

The Einstein relation in two dimensions

$$D = \lim_{t \to \infty} \frac{MSD(t)}{4t} \tag{4}$$

relates the lateral mean square displacement $MSD(t)$ to the diffusion coefficient $D$. Lateral displacements were calculated in the $xy$-plane for the PO4 beads (P atoms for AA systems; ROH, respectively hydroxyl oxygen, for cholesterol) and the diffusion coefficient $D$ fitted for lag times between 5 ns and 10 ns. Analysis was restricted to lipids initially within a central circular analysis domain of radius 3 nm.

## Leaflet assignment and flipping rate of cholesterol

For every frame, all cholesterol molecules were assigned to the closest membrane leaflet (distance between the ROH bead, hydroxyl oxygen for AA systems, and PO4 beads, P atoms for AA systems). The cholesterol

assignment was also used to calculate flipping rates. Flips were counted only if the cholesterol molecule stayed for ≥10 ns within the new leaflet.

## Membrane elasticity

The membrane bending modulus $\kappa_b$ was assessed employing four different methods: the spectrum of membrane undulations (Undulation Spectrum Method)[35,38,77,78], the spectrum of lipid orientations (Orientation Spectrum Method)[35,38,77,78], using the probability distribution of local membrane curvatures (Local Fluctuation Method)[9], and the distribution of splay values of neighbouring lipid pairs (Real Space Fluctuations Method)[79,80].

The Undulation Spectrum Method uses the spectrum of the Fourier-transformed membrane surface as defined by the lipid head positions ($z_{\mathbf{q}}$) to calculate the bending modulus $\kappa^u$, the effective tilt modulus $\kappa_\theta$, and the soft-mode divergence $q_c$[38]:

$$\left\langle |z_{\mathrm{q}}|^2 \right\rangle = \frac{1}{1 - (q/q_c)^2}\left( \frac{k_B T}{\kappa_b^u q^4} + \frac{k_B T}{\kappa_\theta q^2} \right) \tag{5}$$

For CG systems, the center of mass of the PO4, GL1, and GL2 beads (PO4, AM1, and AM2 for DPSM) was used for the lipid head position. For AA systems, the lipid head was defined as the center of mass of the headgroup phosphorus atom (P) and the glycerol backbone carbon atom (C2).

The Orientation Spectrum Method uses the spectrum of the lipid director field in Fourier space ($\mathbf{n}(\mathbf{q})$) to obtain the bending modulus $\kappa_b^o$:

$$\left\langle |n_{\mathrm{q}}^{\parallel}|^2 \right\rangle = \frac{1}{1 - (q/q_c)^2}\frac{k_B T}{\kappa_b^o q^2} \tag{6}$$

Lipid directors are defined as vectors connecting lipid head and lipid tail. Heads are defined as above. For CG systems, the center of mass of the last beads of the fatty acyl chains was used for the lipid tail position. For AA systems, the tail was defined as the center of mass between the terminal methyl carbons of the fatty acyl chains.

The 95% confidence intervals for the moduli obtained through Fourier methods were obtained using parametric bootstrapping following the approach by by Ergüder et al.[78]. To estimate the standard errors ($\delta|z_{\mathbf{q}}|^2$) for the spectrum amplitudes ($|z_{\mathbf{q}}|^2$), block averaging was used. Subsequently, 50,000 new spectra were generated by sampling amplitudes from a Gaussian distribution centered at $|z_{\mathbf{q}}|^2$ with a standard deviation of $\delta|z_{\mathbf{q}}|^2$. The confidence intervals were determined by fitting moduli to these generated spectra.

Since Fourier Methods are not applicable for finite bicelle systems, the Local Fluctuation Method (LFM) was employed instead. LFM directly relates local fluctuations in membrane curvature $H$ to the bending modulus $\kappa_b$. Briefly, it explicitly evaluates the membrane-bending energy

$$E_b = \int dA\, 2\kappa_b (H - H_o)^2 \tag{7}$$

within a defined circular area of the membrane on a grid, assuming a vanishing spontaneous membrane curvature $H_o$, and local deviations $(\Delta H_i)^2 = (H_i - \bar{H})^2$ and the degeneracy of states to be independent of the mean curvature $\bar{H}$.

The probabilities of mean curvatures $p(\bar{H}_1)$ and $p(\bar{H}_2)$ are then related by

$$\frac{p(\bar{H}_1)}{p(\bar{H}_2)} = \exp\left( -\frac{2\kappa_b A\left(\bar{H}_1^2 - \bar{H}_2^2\right)}{k_B T} \right) \tag{8}$$

Using the variance $\sigma^2$ of the normally distributed $p(\bar{H})$, the bending modulus $\kappa_b$ is given by:

$$\kappa_b = \frac{k_B T}{4A} \cdot \frac{1}{\sigma^2} \qquad (9)$$

Here no lipid tilt modulus is included, values for the bending elasticity have thus to be considered as an upper bound for the true elasticity.

For a comparative analysis of bicelle and infinite membrane systems we chose a circular analysis domain for the LFM with radius 3 nm. Circular patches of this size displayed for all systems a comparative degree of cholesterol asymmetry (see Supplementary Fig. 3) and coupling of cholesterol asymmetry to the local mean curvature (Supplementary Figs 4–8). Importantly, fluctuations of the mean curvature within the circular analysis domains showed very good agreement for infinite periodic bilayers and bicelle systems (Supplementary Figs 4–8). In addition, we compared (apparent) $\kappa_b$ values derived from curvature fluctuations (LFM) to those from Fourier methods applied to the periodic bilayer systems (see Fig. 3a), revealing an overall good agreement between the different methods.

Additionally, values for the bending modulus were compared to those obtained using the Real Space Fluctuations Method (RSF)[79,80]. In RSF, the distribution of splay values $S = \nabla \mathbf{n} - \nabla \mathbf{N}$ between pairs of neighbouring lipids is used to calculate the bending modulus $\kappa_b^R$. The lipid directors $\mathbf{n}$ are defined as above for CG systems (for cholesterol the R1, respectively, R5 beads are used). For AA systems, the center of mass of the three terminal tail carbons of both chains are used as tails and the heads are defined as the center of mass of the headgroup phosphorous atom (P), the glycerol backbone atom (C2) and the first three tail carbon atoms (C21, C22, C23, C31, C32, C33). For cholesterol, the director connects the C3 and C17 ring-carbon atoms (atom names taken from CHARMM nomenclature).

For larger systems and hence larger undulations it was proposed before to use a local instantaneous surface instead of the averaged surface employed in the original publications[81]. Here, the local instantaneous normal $\mathbf{N}$ was calculated as described for the thickness calculation, using the center of mass of the C1A and C1B beads (R1 bead for cholesterol) for the surface definition in CG systems. For AA systems, the center of mass of the first three atoms of the tail carbons (C21, C22, C23, C31, C32, C33) was used (C3 atom for cholesterol). For all pairs of neighbouring lipids (surface distance <1.1 nm) the splay values were calculated as the covariant derivative of $\mathbf{n} - \mathbf{N}$ orthogonal to the local membrane normal and in the direction that connects the surface atoms. For each pair of lipid types $i$ and $j$ the distribution of splay values is described by the Boltzmann distribution $P(S)$:

$$P(S) = C \exp\left(-\frac{\chi_{ij} S^2 A_l}{2kBT}\right) \qquad (10)$$

The splay moduli $\chi_{ij}$ for lipid species pairs $ij$ are obtained from a quadratic fit to $-2k_B T \ln(P(S))/A_l$, where $A_l$ is the area per lipid. Monolayer bending moduli $\kappa_m^R$ are calculated from the splay moduli $\chi_{ij}$:

$$\frac{1}{\kappa_m^R} = \frac{1}{\phi} \sum_{i,j} \frac{\phi_{ij}}{\chi_{ij}}$$

$\phi_{ij}$ is the number of neighbour pairs of type $ij$ and $\phi = \sum_{i,j} \phi_{ij}$. The bilayer bending modulus $\kappa_b^R$ is the sum of the monolayer bending moduli $\kappa_m^R$.

## Reporting summary

Further information on research design is available in the Nature Portfolio Reporting Summary linked to this article.

## Data availability

All simulation models, input files and structures are available for download from Zenodo [https://doi.org/10.5281/zenodo.10067080][82]. The source data underlying Figs. 1b-d, 2a-f, 3a-c, 5 and Supplementary Figs 2, 3a, b, 4a-d, 5a-d, 6a-d, 7a-f, 8a-f, 9–11 are available at Figshare [https://doi.org/10.6084/m9.figshare.24518158][83].

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

## Acknowledgements

The authors gratefully acknowledge the compute resources and support provided by the Erlangen Regional Computing Center (RRZE) and the Erlangen National High Performance Computing Center (NHR@FAU). The project was partly supported by the German Science Foundation (DFG) within the SFB1027, *Physical Modeling of Non-Equilibrium Processes in Biological Systems* (project C6, to R.A.B.).

## Author contributions

M.P. and M.F.W.T. performed simulations and data analysis, R.A.B. designed the study, all wrote the manuscript.

## Funding

## Competing interests

The authors declare no competing interests.
