## [Peer Review File · Nature Communications]

Non-Universal Impact of Cholesterol on Membranes: Mobility, Curvature Sensing, and ElasticityReviewers' Comments:

Reviewer #1:

Remarks to the Author:

The authors provide an insightful exploration of the impact of cholesterol on the collective properties of lipid membranes, focusing on the observed contradiction in experimental results for unsaturated bilayers, particularly DOPC lipids. Through the application of coarse-grained and all-atom MD simulations, the authors investigate the influence of cholesterol presence, distribution, and dynamics on various membrane compositions. Their findings suggest that the enhanced cholesterol flip-flopping and lateral distribution in unsaturated bilayers compensate for the expected stiffening effect by favoring negatively curved regions. This work significantly contributes to our understanding of membrane dynamics and provides an interesting perspective on the apparent experimental discrepancy. However, to strengthen the manuscript, address missing information, and enhance clarity, the following revisions are requested:

Validation of Conclusions and Explanation of Discrepancy:

In order to validate their conclusions and explain the discrepancy observed with Chakraborty et al., the authors should also perform an analysis using the fluctuation-based method (RSF-MD) employed by Chakraborty et al. This analysis should be included in the main manuscript and comprehensively discussed to elucidate the origin of the discrepancy.

If flip-flips are indeed responsible for the observed softening in DOPC membranes, it indicates that the softening is a result of non-equilibrium effects. Consequently, these effects are primarily visible in fluctuation methods, where they are expected to occur predominantly at smaller wave lengths characterized by high local curvature. However, at larger wave lengths with lower local curvature, the influence of the constraint in leaflet area likely becomes less significant, leading to the disappearance of these non-equilibrium effects. To provide a more comprehensive understanding, the authors should also consider studying these membranes using a "ground state" method for curved membranes, at least at the coarse-grained level. Although the analytical approximations underlying buckled membrane methods are inaccurate for membrane mixtures, as correctly noted by the authors in the supplementary information (SI), alternative model-free methods have been proposed for calculating the bending modulus (e.g., see <https://journals.aps.org/prl/abstract/10.1103/PhysRevLett.117.188102>).

Biological Relevance of Dynamic Softening/Stiffening:

If the observed effect is solely attributed to flip-flops, it represents a non-equilibrium phenomenon observable at smaller time scales (and wave lengths) in fluctuation methods. In this context, it is appropriate to refer to it as an apparent dynamic stiffening rather than actual membrane stiffening. To provide a comprehensive understanding of the biological relevance of this dynamic softening/stiffening, the authors should explore its implications on the free energy landscape of curvature-generating proteins in vivo. A detailed discussion within the context of cellular processes, such as budding, would greatly enhance the manuscript.

Importance of Experiments with Disabled Flip-Flopping and Lateral Displacement:

The experiments conducted by the authors, such as disabling flip-flopping using repulsive forces and inhibiting lateral displacement of cholesterol with flat-bottom potentials, play a crucial role in validating the main conclusions of the study. It is essential to give these experiments more prominence in the main manuscript rather than relegating them to the supplementary information (SI). Furthermore, it is recommended to separately analyze the effects of flip-flopping and lateral displacement in the experiments, and present the results in a dedicated figure within the main manuscript.

Additional Remarks:

- The figures and text of the paper can be confusing. There is a lot of similar data with different variables being shown at once, making it hard to keep track of what is going on and what is important. I would recommend rewriting parts of the manuscript and figures with this in mind. See below and minor remarks for examples, but this goes for the entire manuscript.

- Fig 2H. "significantly enhanced for lipids with unsaturated acyl chains". But not for AA DOPC? Seems to fit perfectly with DPPC and DPSM. I did not find any mention of the AA observation in the text.

- Fig 3

The coloring of the cholesterol does not seem to make sense with how its defined in the caption. For C, the lower leaflet in a positively curved bilayer has negative curvature and therefore cholesterol should flip into this leaflet (which it does according to the cholesterol density/count in the figure), but the red color indicates that it leaves the lower leaflet. The opposite occurs for D. A leaflet-respective coloring, where red means joining the respective leaflet and orange means leaving the respective leaflet seems to make sense (and would also be easier to understand in my opinion).

In general, what this figure tries to tell could be shown using only the upper leaflets (i.e. A and B), and would improve clarity.

A schematic/snapshot of the mechanism (e.g. side view of curved vs flat bilayer, showing cholesterol) might also be more clear. This is basically to emphasize the main conclusion/mechanic of the paper.

Minor remarks:

- Fig 1

"with and without cholesterol molecules", order of snapshots is "without and with" cholesterol. Perhaps rearrange either text or figure?

What do the orange and cyan symbols represent? Not evident from the caption (except that they are AA results). Also a little unclear in the figure as they are covered by other symbols and not connected by lines (fig C,E,F)

"CG bicelles as circles and infinite, periodic bilayers as squares" -> e.g. "CG bicelles as circles; infinite, periodic bilayer as squares".

- Fig 2G,H

Perhaps add "(#upper-#lower)" to the "Chol asymmetry" axes, to make the figure more clear on its own.

- Fig 3

Add headers to the top and side of the figure, signifying which column represents positive (i.e. A,C) and negative (B,D) curvature, and which row represents the upper (A,B) and lower (C,D) leaflet. Otherwise readers will have to continuously switch between caption and figure to determine what is what. However, also see the major remark regarding this figure.

Reviewer #2:

Remarks to the Author:

This MS uses simulations to address the important effect of cholesterol on lipid membranes. Before reference 16 (BTW, this MS cites the wrong journal) it was believed that cholesterol universally stiffened membranes. (Ref. 16 was the first to declare, even, like this MS, in its title, that the effect is non-universal – a bit of history that is not transparent in this MS.) Since the simulations in this MS agree that the effect of cholesterol is non-universal, what's new? The answer is that this MS reports a mechanism to understand why the effect is non-universal, and that had not been done convincingly until very recently. Namely, that redistribution of cholesterol coupled to curvature leads to a softening that competes with the stiffening that is well understood to accompany thickening due to the condensing effect. The simulations in this MS provide evidence that the requisite cholesterol redistribution is greater for DOPC than for DPPC.

The authors of this MS may be unaware that there is a very recent paper that explains in broadly similar terms how mixtures soften membranes. (Phys. Rev. E 107, 054403 (2023)). This paper only

mentions cholesterol at the end, but it might be appropriate that it be referred to.

The end of the introduction brings up biological significance. The authors might like to refer to a recent paper on the RBC that resonates with this paragraph. Himbert et al., PLOS ONE
<https://doi.org/10.1371/journal.pone.0269619>

A likely concern is that this paper shows cholesterol having a fairly large decrease in stiffness in DOPC, compared to essentially no change experimentally, and a rather modest increase in DPPC, compared to a much larger increase experimentally. The MS glosses over this. I think it would be better to admit this and then say that it is the contrast in the effect on the two lipid bilayers that is important. Quite possibly, flip-flop could be faster and more coupled to curvature due to small infelicities in the force fields and that just tilts the competition to more softening in all bilayers with mixtures of lipids with different spontaneous curvatures.

Thank you for handling our manuscript. We are thankful to the appreciative and supportive assessment of our manuscript by the Reviewers and their constructive comments. In the attached revision, we have addressed all Reviewer comments, as detailed below (main changes in the revised manuscript marked in red).

The main changes to the manuscript are as follows:

- We performed additional microsecond all-atom MD simulations of DOPC at 0% and 40% cholesterol content, at 298K and 310K (before only at 320K), and of DPPC at 330K. The additional simulations show a moderate increase of the DOPC bending modulus at ambient temperature (Fig. 3b, new), in agreement with experiment, and a decrease of bending rigidity at 320K. The DOPC simulations thus support the notion of a diffusive softening of unsaturated membranes by cholesterol. In turn, the rigidity of DPPC upon cholesterol addition is four-fold increased at all-atom resolution. This dramatic stiffening of DPPC is connected to a phase transition to the liquid-ordered (Lo) phase (lower rigidity in coarse-grained simulations; these do not fully grasp the phase transition).
- We added simulations of very large systems (55nm and 110nm length scale, up to 37,000 lipids, coarse-grained resolution) to investigate a possible length scale-dependency of the change in bending rigidity upon cholesterol addition. The analysis shows that the cholesterol-induced softening of DOPC does not depend on the wavelength up to 110nm (Fig. 3c, new).
- We improved the fitting and error analysis for the bending modulus employing parametric bootstrapping (details in Supplementary Methods).
- We added a new Figure (Fig. 5) that shows the effect of cholesterol dynamics on the membrane bending modulus of DPPC and DOPC lipid bilayers: The structural integration of cholesterol stiffens both the saturated DPPC and the double-unsaturated DOPC. Both flipping of cholesterol between the leaflets and lateral diffusion soften the membrane, for DOPC even below the rigidity of the cholesterol-free membrane.

Reviewer #1 (Remarks to the Author):

(R1.0) The authors provide an insightful exploration of the impact of cholesterol on the collective properties of lipid membranes, focusing on the observed contradiction in experimental results for unsaturated bilayers, particularly DOPC lipids. Through the application of coarse-grained and all-atom MD simulations, the authors investigate the influence of cholesterol presence, distribution, and dynamics on various membrane compositions. Their findings suggest that the enhanced cholesterol flip-flopping and lateral distribution in unsaturated bilayers compensate for the expected stiffening effect by favoring negatively curved regions. This work significantly contributes to our understanding of membrane dynamics and provides an interesting perspective on the apparent experimental discrepancy.

Reply: We thank the Reviewer for the positive feedback.

However, to strengthen the manuscript, address missing information, and enhance clarity, the following revisions are requested:

(R1.1) Validation of Conclusions and Explanation of Discrepancy:

In order to validate their conclusions and explain the discrepancy observed with Chakraborty *et al.*, the authors should also perform an analysis using the fluctuation-based method (RSF-MD) employed by Chakraborty *et al.* This analysis should be included in the main manuscript and comprehensively discussed to elucidate the origin of the discrepancy.

Reply: We agree and included a comparison of RSF results with those of Fourier-based methods (added a paragraph in Discussion Section, and added results for RSF analysis for all systems in Supplementary Table 4). We validated our RSF implementation on a simulation of a DOPC bilayer (with/without cholesterol) of the same size (200 lipids) and parameters (298K) as reported in Chakraborty *et al.* (PNAS 117, 21896-21905, 2020). Similar values for the bending modulus were obtained (comparison to values in Chakraborty *et al.*, and values analyzed employing the RSF tool of Allolio *et al.*, Chem. Phys. 514, 31-43, 2018):

0% Chol: 17.6 kT (our work) – 18.3 kT (Chakraborty *et al.*) – 22.4 kT (RSF tool, Allolio *et al.*)
40% Chol: 60.7 kT (our work) – 52.1 kT (Chakraborty *et al.*) – 61.3 kT (RSF tool, Allolio *et al.*)

The deviations of the RSF-based bending moduli from those determined in Fourier-based analysis of membrane undulations increase with increasing cholesterol concentration. However, already for pure DOPC, the bending modulus was reported by the developers to be 35% lower as compared to Fourier-based methods (Doktorova *et al.*, PCCP 19, 16806-16818, 2017). At high cholesterol concentration, the bending moduli using RSF are about twofold larger than those obtained from Fourier-based undulation analysis. We note that the RSF results of substantial stiffening of DOPC membranes upon addition of cholesterol at 320K contradict the observation of increased surface fluctuations.

(R1.2) If flip-flops are indeed responsible for the observed softening in DOPC membranes, it indicates that the softening is a result of non-equilibrium effects. Consequently, these effects are primarily visible in fluctuation methods, where they are expected to occur predominantly at smaller wave lengths characterized by high local curvature. However, at larger wave lengths with lower local curvature, the influence of the constraint in leaflet area likely becomes less significant, leading to the disappearance of these non-equilibrium effects. To provide a more comprehensive understanding, the authors should also consider studying these membranes using a "ground state" method for curved membranes, at least at the coarse-grained level. Although the analytical approximations underlying buckled membrane methods are inaccurate for membrane mixtures, as correctly noted by the authors in the supplementary information (SI), alternative model-free methods have been proposed for calculating the bending modulus (e.g., see <https://journals.aps.org/prl/abstract/10.1103/PhysRevLett.117.188102>).

Reply: We thank the Reviewer for this comment. Both cholesterol flip-flops and lateral cholesterol diffusion contribute to the softening observed in DOPC membranes (at 320K, see also below). To investigate in more detail a possible length scale-dependency of the cholesterol-induced softening, we performed additional simulations of four-fold (55nm x 55nm) and 16-fold (110nm x 110nm) increased DOPC systems at 0% and 40% cholesterol content (coarse-grained resolution). The cholesterol-induced softening was found to be independent of the wavelength up to the investigated length of 110nm (added Figure 3c). A similar softening was recently reported by Kapp *et al.* (Phys Rev E 107, 054403, 2023) for a mixture of phosphatidylcholine with phosphatidylethanolamine, employing kinetic relaxation of GUVs and a Helfrich energy density modified by inclusion of lipids of different spontaneous curvatures (see also comment R2.1 below). Here we show that the effects of diffusional softening by cholesterol may even outweigh the effects of cholesterol-induced membrane thickening and lipid condensation on membrane rigidity.

We note that we observed artificial membrane deformations for systems >50nm employing standard settings. These are caused by neighbourlist artefacts (Kim *et al.* 10.26434/chemrxiv-2023-zbj6j, 2023) as well as by specific LINCS settings combined with cholesterol. We validated the chosen settings by analysis of possible imbalances in the pressure tensor (added Supplementary Figure 10).

In addition to the bending modulus, we now also determined the lipid tilt moduli from the spectra of lipid heights as these soften the membrane at large q values / short wavelengths (added to Supplementary Table 4). Interestingly, the tilt moduli consistently increase upon cholesterol addition, independent of the change in bending modulus. This result is consistent with the observed lipid condensing and increased acyl chain ordering effects of cholesterol (added to Results Section).

We are reluctant to apply "ground state" methods, i.e. methods enforcing specific membrane curvatures, as these limit the free dynamics of the system. E.g., a recent paper that appeared during revision (Fiorin *et al.*, PNAS Nexus 2, 1-12, 2023) employing external forces to enforce sinusoidal shapes of the studied membrane reported a similar degree of stiffening of cholesterol on DOPC at 300K as observed here. However, the authors concluded that the "mechanism of curvature generation is unchanged upon the addition of cholesterol" below a length scale of 10nm. This is at clear variance with our results from unbiased MD simulations. In addition, the authors did not observe local cholesterol enrichment, again at variance with our results (included in Discussion Section).

(R1.3) Biological Relevance of Dynamic Softening/Stiffening:

If the observed effect is solely attributed to flip-flops, it represents a non-equilibrium phenomenon observable at smaller time scales (and wave lengths) in fluctuation methods. In this context, it is appropriate to refer to it as an apparent dynamic stiffening rather than actual membrane stiffening. To provide a comprehensive understanding of the biological relevance of this dynamic softening/stiffening, the authors should explore its implications on the free energy landscape of curvature-generating proteins *in vivo*. A detailed discussion within the context of cellular processes, such as budding, would greatly enhance the manuscript.

Reply: We thank the Reviewer for the comment and suggestion. The softening observed for di-unsaturated DOPC (at 320K) is attributed to cholesterol flip-flops and enhanced cholesterol (lateral) diffusion. We observed this effect on length scales ranging from 5nm (membrane thickness) to 110nm (see also R1.2), a length scale dependency could not be seen. We agree that the LFM method (by construction) rather yields *apparent* bending moduli at smaller length scales. However, the obtained values are in good agreement with those values obtained from analysis of surface undulations at membrane system sizes between 20nm-110nm.

Taken together, the observed softening for DOPC at 320K, slight stiffening at 298K, and strong stiffening of DPPC membranes likely represent actual properties of the investigated membranes, the determined bending moduli determined from undulation analysis should rather be considered as observables of the respective membranes.

Our results suggest that local curvature induction by transmembrane proteins or associated proteins may be supported by local redistribution of cholesterol, thereby releasing membrane stress. Indeed, we observed before asymmetrically distributed cholesterol between the membrane leaflets in vicinity of a curvature-inducing voltage-sensing potassium channel (Pöhl *et al.* JCTC, 19, 1908-1921, 2023). This effect is possibly amplified by the reported enrichment of polyunsaturated lipids near membrane proteins (e.g. Corradi *et al.*, ACS Cent. Sci. 4, 709-717, 2018). Following the suggestion of the Reviewer, we added now a paragraph to the Discussion Section on the role of cholesterol for curvature-generating proteins (PIEZO1 channel), and for membrane remodeling processes (influenza virus infection of cells, clathrin-mediated endocytosis). Experimental data on these systems are in support of the outstanding role of the negative spontaneous curvature of cholesterol and the suggested cholesterol-induced membrane softening.

(R1.4) Importance of Experiments with Disabled Flip-Flopping and Lateral Displacement:

The experiments conducted by the authors, such as disabling flip-flopping using repulsive forces and inhibiting lateral displacement of cholesterol with flat-bottom potentials, play a crucial role in validating the main conclusions of the study. It is essential to give these experiments more prominence in the main manuscript rather than relegating them to the supplementary information (SI). Furthermore, it is recommended to separately analyze the effects of flip-flopping and lateral displacement in the experiments, and present the results in a dedicated figure within the main manuscript.

Reply: We thank the Reviewer for the suggestion. We have included the results on systems including artificial restraints into the main manuscript, presented in Figure 5 in the Results Section, together with a corresponding paragraph.

(R1.5) Additional Remarks:

- The figures and text of the paper can be confusing. There is a lot of similar data with different variables being shown at once, making it hard to keep track of what is going on and what is important. I would recommend rewriting parts of the manuscript and figures with this in mind. See below and minor remarks for examples, but this goes for the entire manuscript.

Reply: We now split the original Figure 1 into two figures, Figure 1 focusing on general membrane characteristics and Figure 3 (new) with a focus on the membrane bending modulus. Additionally, we re-wrote larger parts of the manuscript.

(R1-6) Fig 2H. "significantly enhanced for lipids with unsaturated acyl chains". But not for AA DOPC? Seems to fit perfectly with DPPC and DPSM. I did not find any mention of the AA observation in the text.

Reply: We re-structured Figure 2, including now only data from all-atom MD simulations for DPPC and DOPC to avoid confusion and strengthen the focus.

(R1.7) Fig 3. The coloring of the cholesterol does not seem to make sense with how its defined in the caption. For C, the lower leaflet in a positively curved bilayer has negative curvature and therefore cholesterol should flip into this leaflet (which it does according to the cholesterol density/count in the figure), but the red color indicates that it leaves the lower leaflet. The opposite occurs for D. A leaflet-respective coloring, where red means joining the respective leaflet and orange means leaving the respective leaflet seems to make sense (and would also be easier to understand in my opinion). In general, what this figure tries to tell could be shown using only the upper leaflets (i.e. A and B), and would improve clarity. A schematic/snapshot of the mechanism (e.g. side view of curved vs flat bilayer, showing cholesterol) might also be more clear. This is basically to emphasize the main conclusion/mechanic of the paper.

Reply: We agree and adapted Figure 3 (now Figure 4). To improve clarity, we chose one color for cholesterol molecules leaving the central domain within both leaflets, and one for those diffusing and flipping into the central domain of the respective leaflets. Additionally, we included sketches of the respective curved bicelles.

(R1.8) Minor remarks:

- Fig 1 "with and without cholesterol molecules", order of snapshots is "without and with" cholesterol. Perhaps rearrange either text or figure?

Reply: Corrected in caption, and added legend to Figure.

(R1.9) What do the orange and cyan symbols represent? Not evident from the caption (except that they are AA results). Also a little unclear in the figure as they are covered by other symbols and not connected by lines (fig C,E,F) "CG bicelles as circles and infinite, periodic bilayers as squares" -> e.g. "CG bicelles as circles; infinite, periodic bilayer as squares".

Reply: We improved the clarity of Figure 1 (and new Figure 3, see also above).

(R1.10) Fig 2G,H Perhaps add "(#upper-#lower)" to the "Chol asymmetry" axes, to make the figure more clear on its own.

Reply: We added the cholesterol asymmetry to Figure 2 e,f, avoiding the before provided absolute numbers of cholesterol molecules. The new Figure S11 complements Figure 2 by the local cholesterol asymmetry as a function of distance from grid points of defined curvature.

(R1.11) Fig 3 Add headers to the top and side of the figure, signifying which column represents positive (i.e. A,C) and negative (B,D) curvature, and which row represents the upper (A,B) and lower (C,D) leaflet. Otherwise readers will have to continuously switch between caption and figure to determine what is what. However, also see the major remark regarding this figure.

Reply: See above. Sketches were added to improve clarity, colors adapted, and headers added.

Reviewer #2 (Remarks to the Author):

(R2.0) This MS uses simulations to address the important effect of cholesterol on lipid membranes. Before reference 16 (BTW, this MS cites the wrong journal) it was believed that cholesterol universally stiffened membranes. (Ref. 16 was the first to declare, even, like this MS, in its title, that the effect is non-universal – a bit of history that is not transparent in this MS.) Since the simulations in this MS agree that the effect of cholesterol is non-universal, what's new? The answer is that this MS reports a mechanism to understand why the effect is non-universal, and that had not been done convincingly until very recently. Namely, that redistribution of cholesterol coupled to curvature leads to a softening that competes with the stiffening that is well understood to accompany thickening due to the condensing effect. The simulations in this MS provide evidence that the requisite cholesterol redistribution is greater for DOPC than for DPPC.

Reply: We thank the Reviewer for the positive feedback and appreciate the recognition of our manuscript's contribution in elucidating the non-universal effect of cholesterol on membrane rigidity, particularly through the novel mechanism of cholesterol redistribution coupled to curvature.

We have corrected the reference (Ref. 16), and added it as a reference to the non-universal effect of cholesterol on membrane rigidity.

(R2.1) The authors of this MS may be unaware that there is a very recent paper that explains in broadly similar terms how mixtures soften membranes. (Phys. Rev. E 107, 054403 (2023)). This paper only mentions cholesterol at the end, but it might be appropriate that it be referred to.

Reply: We thank the Reviewer for pointing out this paper to us. The paper introduced the idea of diffusional softening on giant unilamellar vesicles and showed that differences in the spontaneous curvature of lipids may result in softening of membranes that is connected to lateral lipid diffusion (Ref.46). We refer to this paper in the Discussion Section.

(R2.2) The end of the introduction brings up biological significance. The authors might like to refer to a recent paper on the RBC that resonates with this paragraph. Himbert et al., PLOS ONE <https://doi.org/10.1371/journal.pone.0269619>

Reply: We thank the Reviewer for pointing out this work to us. We added a paragraph to the Discussion Section. The paper reports a very low bending modulus of only 4-6 kT for RBC membranes. This is even more puzzling as an anomalous stiffening was observed for cholesterol-free asymmetric lipid compositions (Lu *et al.*, Soft Matter 12, 7521-

7528, 2016; Frewein et al., Biophys. J. 122, 2445-2455, 2023; Ref. 51,52). Possibly, cholesterol in combination with highly asymmetric lipid compositions (see Doktorova *et al.* bioRxiv 10.1101/2023.07.30.551157, 2023) contributes to the softening.

(R2.3) A likely concern is that this paper shows cholesterol having a fairly large decrease in stiffness in DOPC, compared to essentially no change experimentally, and a rather modest increase in DPPC, compared to a much larger increase experimentally. The MS glosses over this. I think it would be better to admit this and then say that it is the contrast in the effect on the two lipid bilayers that is important. Quite possibly, flip-flop could be faster and more coupled to curvature due to small infelicities in the force fields and that just tilts the competition to more softening in all bilayers with mixtures of lipids with different spontaneous curvatures.

Reply: We thank the Reviewer for this comment.

All simulations in the original version were performed at 320K, the default for coarse-grained simulations employing the Martini 2 forcefield. At this temperature, and as noted by the Reviewer, we observed a consistent softening of DOPC membranes upon addition of cholesterol, both at coarse-grained and at all-atom resolution. In contrast, the coarse-grained DPPC simulations showed a modest increase in the bending modulus for DPPC.

Since experiments are typically done at ambient conditions, we now added microsecond all-atom MD simulations of DOPC at 0% and 40% cholesterol content, at 298K and 310K (in addition to 320K), and of DPPC at 330K. We refrained from performing these simulations at CG resolution since coarse-grained forcefields by construction do not accurately reproduce the temperature-dependency of observables. DPPC was studied at the elevated temperature of 330K to avoid transition to the gel phase.

Depending on the temperature, the all-atom simulations of DOPC show a small increase in bending rigidity (+6 kT at 298K, +5 kT at 310K) or decrease (-6 kT at 320K), thus further strengthening the idea of diffusional membrane softening (the diffusion coefficient of cholesterol increases from $0.08 \cdot 10^{-6}$ to $0.15 \cdot 10^{-6}$ cm²/s between 298K and 320K, Table S3).

For DPPC at 330K, a very strong increase in bending modulus is observed (all-atom), from 28 kT (pure DPPC) to 106 kT (40mol% cholesterol), in good agreement with experiment for the fully saturated DMPC (Méléard *et al.*, BJ, 72, 2616-2629, 1997; Pan *et al.*, PRL, 198103, 2008). This increase is explained by transition of the DPPC:Chol system to a liquid-ordered phase with substantially decreased area per lipid and diffusion, and increased lipid ordering. This phase transition is insufficiently captured by the coarse-grained forcefield.

The manuscript was accordingly adapted to cover these additional results for DOPC and DPPC at different temperatures.

Best regards,
Rainer Böckmann

Reviewers' Comments:

Reviewer #1:

Remarks to the Author:

The authors have addressed all my comments.

Reviewer #2:

Remarks to the Author:

Well revised. I appreciate the comments on page 12 regarding the issue of the real space simulation.